# Immunity against *Lagovirus europaeus* and the Impact of the Immunological Studies on Vaccination

**DOI:** 10.3390/vaccines9030255

**Published:** 2021-03-13

**Authors:** Claudia Müller, Rafał Hrynkiewicz, Dominika Bębnowska, Jaime Maldonado, Massimiliano Baratelli, Bernd Köllner, Paulina Niedźwiedzka-Rystwej

**Affiliations:** 1Department of Experimental Animal Facilities and Biorisk Management, Friedrich-Loeffler-Institute, 17493 Greifswald-Insel Riems, Germany; claudia.mueller@fli.de; 2Institute of Biology, University of Szczecin, Felczaka 3c, 71-412 Szczecin, Poland; rafal.hrynkiewicz@usz.edu.pl (R.H.); dominika.bebnowska@usz.edu.pl (D.B.); 3HIPRA, 17170 Amer, Spain; jaime.maldonado@hipra.com (J.M.); massimiliano.baratelli@hipra.com (M.B.); 4Institute of Immunology, Friedrich-Loeffler-Institute, 17493 Greifswald-Insel Riems, Germany

**Keywords:** *Lagovirus europaeus*, hemorrhagic fever, RHDV, rabbit, vaccination

## Abstract

In the early 1980s, a highly contagious viral hemorrhagic fever in rabbits (*Oryctolagus cuniculus*) emerged, causing a very high rate of mortality in these animals. Since the initial occurrence of the rabbit hemorrhagic disease virus (RHDV), several hundred million rabbits have died after infection. The emergence of genetically-different virus variants (RHDV GI.1 and GI.2) indicated the very high variability of RHDV. Moreover, with these variants, the host range broadened to hare species (*Lepus*). The circulation of RHDV genotypes displays different virulences and a limited induction of cross-protective immunity. Interestingly, juvenile rabbits (<9 weeks of age) with an immature immune system display a general resistance to RHDV GI.1, and a limited resistance to RHDV GI.2 strains, whereas less than 3% of adult rabbits survive an infection by either RHDV GI.1. or GI.2. Several not-yet fully understood phenomena characterize the RHD. A very low infection dose followed by an extremely rapid viral replication could be simplified to the induction of a disseminated intravascular coagulopathy (DIC), a severe loss of lymphocytes—especially T-cells—and death within 36 to 72 h post infection. On the other hand, in animals surviving the infection or after vaccination, very high titers of RHDV-neutralizing antibodies were induced. Several studies have been conducted in order to deepen the knowledge about the virus’ genetics, epidemiology, RHDV-induced pathology, and the anti-RHDV immune responses of rabbits in order to understand the phenomenon of the juvenile resistance to this virus. Moreover, several approaches have been used to produce efficient vaccines in order to prevent an infection with RHDV. In this review, we discuss the current knowledge about anti-RHDV resistance and immunity, RHDV vaccination, and the further need to establish rationally-based RHDV vaccines.

## 1. Introduction

Interactions between (viral) pathogens and their hosts certainly play a key role in the development of biological diversity, following an ever-ongoing ‘arms race’ in which pathogen virulence and host resistance mechanisms co-evolve with each other, leading to a dynamic equilibrium. The Calicivirus that causes rabbit haemorrhagic disease probably evolved from low-virulence strains, and is still a huge problem in the world today. Despite the fact that the disease has been circulating in the environment for about 30 years, it still has a high mortality rate [1]. The observed innate resistance of young rabbits to infections, coupled with the high fertility of the species, causes the population to recover quickly. The evolution of the resistance of rabbits to infections was also recorded [2], which together with the phenomenon of cross-protection after infection with non-pathogenic Australian rabbit calicivirus resulted in the reduction of virulence of the virus [3,4]. In turn, the virus, in order to maintain its virulence at a high level, shows the ability to use various vectors to spread over long distances, and cases of breaking the species barrier have been observed [1,3].

In the last quarter of 1983, an extremely contagious disease for European rabbits (*Oryctolagus cuniculus*) emerged, killing—in the following 12 months after the first outbreak—over 140 million domestic rabbits in China [5]. The disease spread in this short time over an area of about 500,000 km^2^. Due to the observed pathology, with pronounced damage in the liver and the severe dysregulation of the coagulation system, the disease was described as rabbit hemorrhagic disease (RHD) [5,6]. In the following decade, the virus spread throughout the globe, and threatens the state economies that are based on the rabbit industry [5,7,8,9,10,11,12,13,14,15,16,17]. RHDV had also a negative impact in Mediterranean ecosystems, were rabbits are a food base for endangered endemic species, including the Iberian eagle (*Aquila adalberti*) and Iberian lynx (*Lynx pardinus*) [18,19]. In contrast, the killing of 90% of the ‘pest’ rabbit population in Australia in very short time by RHDV was a desired impact on the ecosystem there [11,20,21,22].

The causative agent was identified as a Calicivirus [23,24,25,26,27,28]. Until 2010, all RHD outbreaks were caused by the so-called ‘classical’ RHDV (GI.1). In 2010 [29], in France, a new RHDV strain emerged, designated as RHDV2 (GI.2), which not only infects *O. cuniculus* but also hare species *Lepus capensis mediterraneus* [30], *L. corsicanus* [31], *L. europaeus* [32,33], and *L. timidus* [34]. In the 1990s, the classic RHDV GI.1 strains were detected in *Lepus granatensis* [34]. These infections in Lepus species indicate that both RHDV GI.1 and GI.2 have the ability to break the species barrier and infect other species besides *O. cuniculus* [34,35]. In 2020, there were new RHDV GI.2 epidemics in the Southwestern United States and Mexico, with reported mortality rates of over 90% [36] in black-tailed jackrabbits (*Lepus californicus*), antelope jackrabbits (*Lepus alleni*), desert cottontails (*Sylvilagus audubonii*), and mountain cottontails (*Sylvilagus nuttallii*). There are concerns that RHDV infection could significantly adversely affect these already endangered North American species of lagomorphs [37]. However, the pathology of RHD is similar, and is characterized by a severe clinical course and high mortality rates from 90 to 100% as a result of progressive multi-organ failure resulting from the development of DIC syndrome [38].

The clinical course of RHD appears in three forms: subacute, acute, and chronic [39,40], with respiratory, nervous and digestive disorders as a result of pathological changes in the inner organs and pronounced damage in the liver [5,41]. One phenomenon of RHD is the resistance of juvenile rabbits against a RHDV infection until the age of 4 weeks (GІ.2 strains) and 9 weeks (GІ.1 strains), respectively, in contrast to the about 3% of adult rabbits which survive after more or less severe clinical symptoms [6].

The pathology after RHDV infection is characterized by two main processes:(a)Disturbed physiology with the severe destruction of the target organs (liver, spleen, kidney) along with an extremely rapid RHDV replication within 36–72 h post infection, with up to 2^14^ HU related to 10^8^ copies of viral RNA per g of liver tissue [42]. Immunohistological examinations and in-situ hybridization (ISH) studies of infected adult rabbits have demonstrated the presence of RHDV in periportal hepatocytes, and macrophages of the liver, lungs, and spleen [43,44]. The disease progression is correlated to an increased apoptosis of hepatocytes and liver endothelial cells [43]. The disturbed coagulation by increased prothrombin time, increased fibrin degradation, and decreased antithrombin III activity results in micro-thrombi, DIC, and a hypervariable condition in blood coagulation [45]. Liver enzymes significantly increase, whereas antioxidant enzymes decrease within the first 36 h post infection [46]. Respiratory acidosis, hypoglycemia, and increased creatinine kinase activity indicate multi organ dysfunction, which finally results in death.(b)Disturbed immune response with an induction of systemic apoptosis of lymphocytes, and especially T-cells, leading to a nearly complete loss of the immune effectors within 48–72 h after infection. In adult rabbits, the apoptosis of T- and B-cells in the liver and peripheral blood [47,48] with the infiltration of neutrophils [49] ends in a decrease of regulatory T-cells [50] and severe leukopenia before death [51]. After RHDV infection, an increase of pro-inflammatory cytokines in the serum, liver, and spleen of young and adult rabbits [50,52], and also anti-inflammatory cytokines in adult rabbits has been reported [53].

Due to the enormous severity of the disease and the impact on the production of rabbits worldwide, several pharmaceutical companies started to develop a vaccine against this deadly viral disease (Table 1). Unfortunately, until now, no cell culture system existed which allows an in-vitro cultivation of native RHDV strains. Therefore, the first vaccines were based on inactivated RHDV prepared from the liver of RHDV-infected rabbits (Table 1) [54,55,56]. Therefore, early after the introduction of such inactivated ‘liver’ vaccines, several recombinant vaccines containing RHDV and the RHDV-2 capsid protein VP1 expressed in different in-vitro expression systems were developed [57,58,59,60,61,62,63]. The vaccines against RHDV were demonstrated to have limited cross protection against RHDV-2, and vice versa; for this reason, the current vaccines contain one or both virus-derived antigens in their formulation [64,65,66,67,68]. To date, two different types of genomic recombination have been described: one in the RHDV VP60 sequence region, and a second between the structural and nonstructural genome regions [64]. Partially non-pathogenic GІ.1 strains were detected in the late 1990s, which provide cross-protection against circulating pathogenic RHDV [69].

Therefore, an in-depth characterization of the humoral and cellular immune mechanisms after RHDV infection comparing the different regulation in juvenile and adult rabbits could provide the knowledge base for rational anti-RHDV vaccine development.

## 2. *Lagovirus Europaeus*—Epidemiological History of the RHD Causative Agent

The origin of RHDV is not fully understood. The pathogenic virus forms may have evolved from avirulent strains which circulated in European rabbits asymptomatically [66,70,71,72,73] for at least 30 years before the first outbreak of RHD in China. A possible common ancestor of both RHDV and Rabbit calicivirus-like viruses from 200 years ago was predicted, which then mutated to the virulent RHDV strains which emerged in 1983 [74]. A second hypothesis is based on spillover infections of caliciviruses found in micromammals living close to wild rabbit populations [6,29,75,76]. Similarities in the clinical course and outcome after infection with European brown hare syndrome virus (EBHSV), first detected in Denmark and Sweden in 1980 in European brown hares (*L. europaeus*) to RHD in rabbits indicated an evolutionary relation. However, the limited sequence homology of about 76% between RHDV and EBHSV VP1 indicated that RHDV did not evolve from EBHSV [77,78,79]. This is proven by the fact that hares which survived an EBHSV infection displayed no cross immunity against RHDV [80].

The first outbreak of RHD in Europe was detected in Italy in 1986 [81] only two years after the epidemics in China and Korea in 1984 [5,82,83]. Within ten years, RHDV became endemic in Europe, resulting in a severe reduction of wild rabbit populations, especially on the Iberian Peninsula [84,85,86]. Furthermore, in domestic rabbits, RHDV caused dramatic losses in Europe as well as in North Africa [9]. In 1988, RHDV reached the American continent, with limited outbreaks in domestic rabbits in Mexico. However, due to the absence of a susceptible wild rabbit population, it was eradicated shortly after [41]. In 2000, in North America, a limited number of RHDV outbreaks was reported, and also in geographically distant regions—such as Cuba, Uruguay, and Reunion Island—RHDV caused losses in domestic rabbits [87,88]. Nowadays, RHDV is endemic in most parts of Europe, Asia, and parts of Africa.

In 1995, despite the rigorous quarantine measures, RHDV escaped from the Wardang Island in Spencer Gulf, South Australia, where the Czech V351 strain of RHDV was introduced as a biocontrol agent in 1991 [11,20,22]. Within two years, RHDV spread all over southern Australia, reducing by up to 95% the unwanted ‘pest species’ European rabbit, which is a major threat to the endemic wildlife [21,89]. A similar impact on the population of non-native European rabbits was seen in New Zealand, where the Czech V351 RHDV strain was illegally introduced [90,91]. A new RHDV strain—designated as RHDV2—emerged in 2010 in France, and spread rapidly across Europe and worldwide soon afterward as well [32].

*Lagovirus europaeus* belongs to the Caliciviridae family, genus *Lagovirus* [92]. Le Pendu et al. [93] renamed the RHDV nomenclature, and the lagovirus strains were divided into two main gene groups associated with RHDV (GI) or with the European brown hare syndrome virus (EBHSV, GII). Both, GI (RHDV) and GII (EBHSV), are closely related with regards to clinical signs, pathological and histopathological alterations, mortality rates, virion morphology, and antigenicity, but cross-species infection and cross-species protection could not be obtained for RHDV, but was found in some cases for RHDV2 [94,95]. Despite the similarities, *L. europaeus* GI and GII represent distinct agents infecting different species, although they cause similar diseases [40,78,80,96,97,98].

Based on genetic variability, the RHDV GI genogroup can be distinguished into 4 genotypes: GI.1, GI.2, GI.3 and GI.4. The GI.1 genotype includes the classic RHDV strains (*Lagovirus europaeus*/GI.1), while the GI.2 genotype was classified as RHDV2 (*Lagovirus europaeus*/GI.2). According to the recently-proposed nomenclature [93], GI.1 can be further divided into different antigenic variants (GI.1 a-d) based on phylogenesis and genetic distances [93]. To date, the best-tested genetic variants of *Lagovirus europaeus* are the GI.1a variants [47,99,100,101,102]. The main features of GI.1, G.I.1a, and G.I.2 are shown in Table 2.

The GI.3 genotype is represented by RCV-E1 (*Lagovirus europaeus*/GI.3), and genotype GI.4 is represented by RCV-A1 (*Lagovirus europaeus*/GI.4) and RCV-E2 (*Lagovirus europaeus*/GI.4d). The strains found in these genotypes are referred to as non-pathogenic rabbit calicivirus-like viruses (RCV-like viruses). Strains from the GI.1 and GI.2 genotypes are known infectious agents responsible for RHD according to the recently proposed nomenclature [93].

The EBHSV can be divided into two groups [103]. These two groups were classified as the GII genogroup, which contains two genotypes: GII.1, which corresponds to EBHSV, and GII.2, which includes non-pathogenic strains of lagoviruses infecting hares, which are called HaCV. Additionally, variants were described among the GII.1 genotype: GII.1a (G1/group A), GII.1b (GI.3/group B), and GII.1c (G2/group B) [93].

*Lagoviridae* are positive-sense single-stranded RNA viruses with a non-enveloped icosahedral capsid. All RHDV virions in all genotypes share a similar structure: the single subunit capsid protein VP1 forms the virion, the minor structural protein VP2 is responsible for stability after the incorporation of viral RNA, and the so-called VPg (viral protein genome-linked) is essential for virus replication [104]. The capsid protein VP1 is the main viral antigen, as it is exposed to the humoral response of the host due to the lack of an envelope. Furthermore, the C-terminal sequence of this protein is exposed on the external surface, and is mostly responsible for the antigenic variability within this virus family [105]. VP1 induces high titres of virus-neutralizing antibodies in surviving hosts, and displays the antigenic variability within this virus family [13,105,106,107]. Because of the high variability between the genotypes, cross protective immunity is induced at a limited level only. This was also found in several studies in which vaccines against RHDV GІ.1 strains provided only limited cross protection, and vaccines against RHDV GІ.2 provided even less [108,109,110,111]. Due to its high variability and recombination ability, the evolution within the RHDV genogroups is ongoing. In addition, it is likely that new strains with different virulence will appear. Whether or not recent vaccines will induce an immunity which will also protect against new emerging strains is at least questionable.

## 3. Host-Pathogen Interaction in Rabbits after Infection with RHDV (*Lagovirus europaeus*/GI)

Infection with RHDV causes very rapid, severe organ destructions in the inner organs, especially the liver. Moreover, an immune pathological process starting with a polyclonal activation of the lymphocytes leads to dramatic losses of these immune effectors, and finally a multi-systemic, multi-organ failure, and the death of infected host [6]. This extremely rapid disease progress is even more impressive given the fact that very low doses are sufficient to infect a naïve host [112].

The virus is transmitted mainly orally, but also by nasal, conjunctival, or parenteral routes, directly between animals and indirectly by insects, which take up the virus from the excretions of infected animals [113,114,115]. There is an extremely short time (24–60 h post infection) between infection with very low doses in peripheral mucosal surfaces and the overwhelming viral load in inner organs’ requests for an initial replication in these peripheral epithelia, and furthermore, for innate immune mechanisms which might be decisive for survival.

An important aspect of the characterization of RHDV infections (*Lagovirus europaeus*/GI) is its inability to cause severe disease in juvenile rabbits younger than 9 weeks (RHDV GІ.1 [113]) or younger than 4 weeks (RHDV GІ.2) [116] compared to adult rabbits. This is especially remarkable in respect to the immature immune system of juvenile rabbits, especially of the B-lymphocyte ontogeny and maturation [117], which request innate immune mechanisms preventing the initial high replication of RHDV, or which regulate adaptive immune mechanisms to block RHDV replication in the inner organs, especially the liver. The regulation towards T-effector cell-mediated immune mechanisms in juvenile rabbits might be similar to rare cases of adult rabbits surviving the RHDV infection, in which a less severe T-cell loss is seen compared to moribund rabbits [102,118]. The survival of adult rabbits seems to be positively correlated to an IFNγ-blocking of viral replication [53,55,119]. The following B-cell response ending up in high titers of RHDV neutralizing antibodies can protect against future reinfection, but is not responsible for survival—as it takes too long—but rather for long term protection [120,121]. The question is whether the innate effector and/or regulatory mechanisms in juveniles and adult rabbits surviving an RHDV infection are different. The reasons for the two different resistance patterns after RHDV infection in resistant juveniles and fully-susceptible adults were investigated based on different suggestions:(a)Missing virus receptor in peripheral mucosal surfaces.Juvenile rabbits do not express, or only express in incomplete formation, one identified attachment factor for RHDV, the so called HBGA (histo-blood group antigens) in the upper respiratory and digestive tract until the age of 9 weeks [122]. However, entry into hepatocytes is not provided by HBGA because they are not expressed at any age on hepatocytes [122]. Moreover, even in juvenile rabbits, RHDV was detected after infection in the liver [123,124]. Finally, immunosuppressed juvenile rabbits develop a full disease state as adults, indicating that immune mechanisms are involved in the juvenile resistance against RHDV [125].(b)Different replication kinetics in peripheral mucosal surfaces.Flies or direct contact transfer RHDV via the oral/oculo-nasal aerogenic route through the upper respiratory tract with an extremely low dose of possibly less than 100 viral particles [112]. However, between 24 and 48 h post-infection, RHDV replicates to very high titers in the liver of adult rabbits, whereas the amount of RHDV in the liver of juveniles remains low [124]. Whether this is due to limited virus replication in peripheral mucosa in juveniles or due to more effective type I or III IFN responses in juveniles compared to adult rabbits is not clear.(c)Different effective innate immune responses.The pathological processes (see above) leading finally to a DIC are described for both juvenile and adult rabbits [124,126]. Protective immune mechanisms preventing severe clinical processes are mostly developed in juvenile rabbits, but rarely in adults, The more robust innate immune response in juvenile rabbits was characterized as one reason for the RHDV resistance in a transcriptome analysis-based study [127]. The involvement of cellular immune mechanisms, and especially of CD8^+^ T-cells in adult rabbits in protective responses were elucidated [111,128].(d)Different regulation of the Th1/Th2 balance in juvenile and adult rabbits.In surviving adult rabbits, an early and strong increase of IFNγ as well as a limited loss of T-cells was determined. The course of the disease is less severe with a reduction of the viral replication in the liver, an increase of CD8^+^ T-cells, and an increase of IFNγ as well as type I IFNs [53]. Similar response patterns are described for juvenile rabbits where macrophages, T- and B-cells increased in the liver and spleen [52,123]. The reasons for the ‘decision’ towards Th1-based immune responses are different in adults and in juveniles. In juveniles, it might be based on their immature B-cell development [117], as the different outcome after RHDV infection is related to B1 cells contributing to innate immune responses, and not B2 cells as part of the adaptive immune response [129].

However, neither the regulation towards immune mechanisms, which provides final protection, nor the cell populations involved in such early regulatory processes are characterized yet. Consequently, whether the regulation at immune checkpoints towards protective Th1 immune mechanisms and the immune cells inducing and regulating these processes are the same in juvenile and in adult rabbits is also unclear.

### 3.1. Innate Immune Mechanisms Involved in the Anti-RHDV Immune Response of Juvenile and Adult Rabbits

In naïve animals which encounter a viral pathogen for the first time, an effective innate immune response is in most cases decisive for survival or severe disease and death. This is provided either by the complete blocking of viral replication, and afterwards clearance at the peripheral epithelia, or by decreasing the viral replication for a certain time to a level that—following adaptive immune mechanisms—can reach a level to clear the viral pathogen.

#### 3.1.1. Pathogen Sensing

The immune response to virus infection is initiated when the PRRs (pathogen recognition receptors) of the host cells recognize specific non-self pathogen-associated molecular patterns (PAMPs) which introduce innate immunity by triggering intracellular signalling events [130,131].

PRRs include, among others, toll-like receptors (TLRs) located on the cell surface, as well as nod-like receptors (NLRs), type-C lectin receptors (CLRs), RIG-I-like receptors (RLRs) and Pyrin-HIN domain name receptors (PYHIN) (Figure 1). PRRs are responsible for the recognition of PAMPs, and moreover DAMPs—damage/danger-associated molecular patterns which occur through infection-induced cell losses. The activation through these sensors initiates the production of cytokines and chemokines in the host organism, which activates NFκB and IRF-3 factor, and finally leads through type І IFN and proinflammatory cytokines (IL-1, IL-2, 1L-6, or IL-12 or TNF-α) to an antiviral status [53,132,133].

In the case of RHDV, the incubation period is 1 to 3 days, i.e., very short, and rabbits usually succumb within 12 to 36 h after the onset of fever (>40 °C). The rapid and fatal course of the disease suggests that the rabbits usually die before mounting an effective adaptive immune response against RHDV infection, and moreover that innate virus replication-blocking immune mechanisms are obviously decisive for survival or death.

Concerning rabbits, the general knowledge about PRR is very limited. Only TLR1 [134], TLR2 [135], TLR3 [136], TLR4 [137], TLR8 [138], and TLR9 [139] (GenBank Accession number: KC349941, NM001082781, NM001082219, DQ250128, AY101394, JX948743, HM448400) have been identified and sequenced until now. Further putative TLR sequences of rabbits have been identified but not yet functionally characterized (TLR5 Acc. Number: HQ874605, TLR6 Acc. Number: KC349945, TLR10 Acc. Number: KC349942, TLR11 Acc. Number: KC349943, TRL12 Acc. Number: KC349944). However, the complete sequence of TLR1 is known [134], but might be of lesser importance in viral infections of rabbits, because it is known to have a more antibacterial effect. TLR3 was identified by recognizing double stranded viral RNA, and was shown to be very diverse among *Oryctolagus cuniculus* [140]. In total, 41 single nucleotide polymorphisms (SNPs) in the sequence of TLR3, and some substitutions, were observed within the LRR motifs in the TLR3 molecule [140]. Only TLR4 was shown to play a role in a RHDV infection. It was activated during RHDV infection through binding of the high mobility group box 1 protein (HMGB1), which is secreted by immune cells during inflammation [141]. The receptors known to recognize single stranded viral RNA, TLR7 and TLR8, were found to be absent or were previously pseudogenised [136] in rabbits. However, it was shown later that TLR8 is a functional gene, but with lower activity than in other mammals [138]. TLR9 is preferentially expressed in immune cell-rich tissues. However, the involvement in immune responses against real bacterial pathogens is not yet shown [139]. In general, the involvement of TLR pathways in the recognition of RHDV and the possible influences between resistant juveniles and fully susceptible adult rabbits are missing, and should be definitely explored further.

Among NLRs, studies are rare; only a few reports are known about the presence of NOD-like receptors in rabbits, for example rNLRX1 in the infection of rabbits with enterohemorrhagic Escherichia coli [142]; nevertheless, these kinds of receptors are also described to take part in bacterial infections more often than viral infections [130]. Interestingly, only one report exists about the diversity of RIG-I-like receptors among leporids (*Oryctolagus, Sylvilagus, Lepus*), which were shown to be involved in immune responses against myxomatosis [143].

#### 3.1.2. Apoptosis and Autophagy in RHDV Infection

The major histopathological lesions found at necropsy are acute hepatitis due to liver cell loss as the result of RHDV-induced apoptosis, and splenomegaly. Haemorrhages and congestions can be seen in several organs—particularly in the lungs, heart, and kidneys—as a result of a massive disseminated intravascular coagulation (DIC), which is usually the cause of death [45].

The process of programmed cell death plays one of the key roles in the pathogenesis of RHD, and affects various cells. Apoptosis in rabbits infected with different strains of the RHD virus was first detected using histopathological methods, i.e., 40 h p.i. in hepatocytes [43]. In addition, the presence of apoptotic cells in RHDV (*Lagovirus europaeus*/GI.1) infection has also been confirmed in macrophages of the abdominal cavity, epithelial cells, lungs, kidneys, heart, spleen, and lymph nodes [43,49]. More detailed studies on apoptosis in immune cells after RHDV virus infection were performed using six RHDV (*Lagovirus europaeus*/GI.1) hemagglutinogenic strains (24V, 1447V/96, 01-04, 237/04, V-412, 05-01) and three non-hemagglutinogenic strains (Rainham, Frankfurt, and Asturias). This showed that apoptosis in the granulocytes and peripheral blood lymphocytes is activated from 4–8 h p.i. in the case of non-hemagglutinogenic strains, and from 12 h p.i. with hemagglutogenic strains [47]. This process intensifies in both cases up to 24–36 h p.i. Apoptosis was more intense in lymphocytes after infection with both hemagglutinogenic and non-hemagglutinogenic RHDV (*Lagovirus europaeus*/GI.1) strains [47,48]. Autophagy is also activated during RHDV (*Lagovirus europaeus*/GI.1) infection. This process weakens as the disease develops, which in turn is accompanied by an increase of programmed cell death. This phenomenon probably shows a protective effect of the organism in response to virus infection. Therefore, further research is necessary in order to understand the process of autophagy in RHDV (*Lagovirus europaeus*/GI.1) infection, and the influence of different virulent RHDV strains [126].

The first study on the phenomenon of the apoptosis of peripheral blood granulocytes and lymphocytes in rabbits infected with RHDVa (*Lagovirus europaeus*/GI.1a) strains—including three hemagglutinogenic strains (Triptis, Hartmannsdorf, Vt97) and two non-hemagglutinogenic strains (9905 RHDVa, Pv97)—showed that, for all of the strains tested, apoptosis is expressed through changes occurring in the percentage share of apoptotic granulocytes and lymphocytes. These changes were recorded at 4–8 h p.i. and 24–36 h p.i. [144]. Moreover, the frequency of apoptotic changes was higher in the case of apoptotic granulocytes, and the average number of these cells remained ten times lower compared to apoptotic lymphocytes [144]. A number of studies show that during RHDVa infection a different immunological picture is observed than in RHDV, and these differences relate primarily to specific cellular and humoral immunity and the process of apoptosis [47,102,145]. These studies show that the biological features of RHDV GI.1 strains (hemagglutination capacity, antigenic variability) have an impact on the immune response.

To date, studies on apoptosis and autophagy in rabbits infected with genotype RHDV GI.2 strains have not been conducted. A better understanding of the different immune processes after infection with different RHDV genotypes in relation to the different outcome of such infections in juvenile versus adult rabbits is certainly a key element for rationally-based vaccine design.

#### 3.1.3. Apoptosis and Autophagy in RHDV Infection

An important aspect to understand the different immune responses in naïve juvenile versus adult rabbits to RHDV infections is based on the characterization of cytokine induction and regulation. Several studies have indicated that cytokines play an important role in modulating the immune response through the activation of peripheral blood leukocytes and their recruitment to the liver (Figure 1). An increase of tumor necrosis factor alpha (TNF-α), interferon alpha (IFN-α), interferon gamma (IFN-γ), transforming growth factor beta (TGF-β), and interleukin-1, -6, -8, and -10 has been observed [50,146]. In the liver, a decrease of hepatocyte growth factor (HGF) expression, which is responsible for liver regeneration, and an increase of the activity of TNF-α, TGF-β, and IL-1β and -6 was measured [46,147,148]. The analysis of cytokine patterns in peripheral blood leukocytes showed a decrease in IL-1β, -2, and -18 activity, and an increased expression of IL-6, -8, -10, TNF-α, and TGF-β genes [53,133]. It was speculated that TNF-β, similarly to TNF-α, which extends the life of infected rabbits by 8-12 h, can act in the same way [149].

Furthermore, the analysis of the gene expression pattern of key elements in the regulation of innate immunity indicated that the different resistance of young rabbits and adults after RHDV GI.1 or GI.2 infection is based on these early recognition and regulation processes [127]. In response to RHDV GI.1, particularly, genes of the major histocompatibility class I complex, as well as interferon-induced genes, were upregulated in juvenile rabbits, accompanied by an increased activity of natural killer cells and CD4^+^ T-cell activation. These phenomena could be important for the resistance of juveniles [127]. In contrast, after RHDV GI.2 infection, these were downregulated, which correlates with the limited resistance of juveniles older than 4 weeks of age. Interestingly, in adult rabbits, MHC (major histocompatibility complex) class II genes were upregulated, indicating an induction of a Th2 immune response known to be unable to protect naïve rabbits because of the rapid disease progress [127].

Possibly, these pathological processes impacting mainly the liver and the cytokine storm may be the direct cause of disseminated intravascular coagulopathy [126,150].

#### 3.1.4. Innate Effector Component and Functions

In the course of RHD, several innate cellular effectors provide the body’s first and primary line of defense in contact with viral pathogens [130].

Neutrophils (polymorphonuclear cells, PMNs) are crucial in innate immune responses due to their cytotoxic, phagocytic and neutrophil extracellular trap forming (NET) activities [151] after the recognition of PAMPs by TLRs. Moreover, neutrophils produce active substances such as lysozyme (LZM), myeloperoxidase (MPO), elastase, esterase, or defensins, which provide anti-viral activities [151,152]. In two studies, the activities of PMNs were measured in the early course of RHD by a variety of parameters (neutrophil phagocytosis, PMN adherence capacity, spontaneous and stimulated metabolic activity, etc.). Most of the parameters indicating functional activation were increased as early as 4–8h p.i. [153]. Interestingly, after infection with *Lagovirus europaeus*/GI.1a, the same parameters decreased, starting at 8–12 h p.i. [101,152], suggesting a different regulation of innate immune mechanisms depending on the genotype of the virus which is probably responsible for the rapid deaths of rabbits in these studies 24–36 h p.i. after RHDVa infection.

Furthermore, during RHDV (*Lagovirus europaeus*/GI.1) infection, a polyclonal hyperactivation in immune cell populations was observed, leading to a recruitment of immature immune cells from the bone marrow within the final stage of RHD before death. Neutrophils display cytoplasmic vacuolization, toxic granules, and Doehl’s bodies as a sign of a systemic dysregulation within the hematopoietic tissues. Macrophages were present in increased amounts, showing the vacuolization of the cytoplasm and phagocytized erythrocytes, lymphocytes, and platelets [146]. In order to characterize the dysregulation of the different functional activities of the innate effector cells, further studies investigating the regulatory gene networks are needed.

The role of natural killer cells which are involved in immediate cytotoxicity against virus infected cells has not been well investigated, yet. In a gene expression study comparing the response of juvenile and adult rabbits to infection with RHDV GI.1 and GI.2 strains, the genes involved in activation and function of NK cells (PTPN22, VAV1, and ARRB2) were downregulated in GI.2.-infected juveniles, while they were upregulated in GI.1-infected juveniles which survived the infection, compared to adults infected with GI.1. [127]. Also, the natural cytotoxicity triggering receptor 3 (NCR3) was elevated. This indicated that, in naïve rabbits, the activation of innate effector cells is crucial for survival.

### 3.2. Antigen Presentation

Virus particles are sensed by pathogen recognition receptors (PRRs) on the surface [154,155,156] or in the endosomes [157] of myeloid cells, mainly dendritic cells (DC). The uptake of such particles activates DC, which afterwards mature. Mature DC process and present peptides derived from viral proteins loaded into major histocompatibility complex (MHC) class I or class II molecules to the T-cell receptor complex of CD8^+^ or CD4^+^ T-cells, respectively. It is worth mentioning that the CD4 gene shows an overall higher divergence in lagomorphs in comparison to primates, with the highest divergence in the D2 domain [158]. The interaction between DC and T-cells leads to an activation and proliferation followed by adaptive effector function via Th1 and/or Th2 responses. Ideally, a Th1 response limits the viral replication by cytotoxic CD8^+^ T-cells which lyse virus-infected cells or abrogate the viral replication through IFNγ-promoted Type I IFN responses. A Th2 response helps B cells to produce virus neutralizing antibodies which act mainly through the blocking of the viral epitopes which are necessary to interact with virus receptors [159,160].

However, in the ‘battle for survival’ between viral pathogens and vertebrate hosts, this response is often counteracted by viral proteins which are able to block the intracellular processing and presentation machinery in DC at different stages [161].

#### 3.2.1. DC’s Interaction with RHDV

Although the immune response of naïve rabbits against RHDV infection is widely studied, only limited data are available on whether the viral replication also happens in myeloid cells (monocytes, macrophages, DC). RHDV VP1 was detected during infection in Kupffer cells in the liver, as well as in circulating monocytes, lymphocytes, and macrophages in the red and white pulp of the spleen, lung macrophages, the glomerular mesangial cells of the kidneys, and lymphocytes in the thymus and lymph nodes [162,163,164]. Whether this indicates an active replication in these cells or whether RHDV VP1 was detected after the uptake of other infected cells is not clear. However, the detection of RHDV in Kupffer cells was connected with signs of apoptosis [43].

#### 3.2.2. Antigen Processing

Obviously, in adult rabbits, the initiated Th2 immune response after infection is counteracted by RHDV through an apoptosis induction in T-cells [6,127]. Whether DC at the peripheral mucosal surfaces or Kupffer cells in the liver are able to process and present RHDV VP1 epitopes is not clear. In juvenile rabbits, an increase of MHC class II early after the infection with RHDV GI.1 was connected with an upregulation of CD4^+^ gene activity, and also with an increased survival rate in contrast to RHDV GI.2-infected rabbits [127]. The extremely rapid pathological progress in the liver normally leads to death before a successful induction of adaptive Th2 immune mechanisms can occur.

However, in the rare cases of the survival of naïve RHDV-infected adult rabbits, an effective CD8^+^ T-cell response protects the liver from severe damage by the secretion of a high level of IFNγ. Moreover, a strong B-cell response provides high titers of RHDV neutralizing antibodies [102,118]. Both indicate that an effective processing and presentation of RHDV VP1 epitopes via MHC I and MHC II by DC occurred.

In contrast to the severe course of RHD in adults, in juvenile rabbits a completely different outcome is seen after infection with RHDV, which is not only indicated by survival but by a complete blockage of the viral replication in the inner organs [6]. An early and very effective type I IFN response might be decisive for this resistance. However, there are also differences in antigen presentation compared to adults, which are moreover different after infection with GI.1 or GI.2 RHDV strains. After GI.1 RHDV infection, the antigen presentation leading to an effective CD8^+^ T-cell response via MHC I as well as the MHC II pathways triggering mainly the B-cell response are upregulated. After infection with GI.2 strains which are able to infect and kill juveniles older than 4 weeks, the MHC II pathways are downregulated. This different activation of antigen presentation pathways might be the reason for the different resistance of juvenile rabbits [127].

### 3.3. Adaptive Immunity

The mechanisms of acquired adaptive immunity evolved in early jawless vertebrates with the appearance of variable lymphocyte receptors (VLR) comprised of leucine-rich-repeat (LRR) segments [165], and further in jawed vertebrates with the immunoglobulin receptor gene family and the recombination activation genes in B- and T-lymphocytes [166]. These effector cells create a second line of defence after the recognition of variable epitopes in pathogen-derived proteins. This second line of defence is activated when the innate immune mechanisms cannot completely clear an infection due to the high number or virulence of the invading pathogens. However, the activation of adaptive immune mechanisms towards a ‘new’ pathogen in naïve hosts needs 1–2 weeks to reach full efficacy. The disadvantage of this late first adaptive response is afterwards turned into the advantage of quicker and more effective response due to immunological memory by circulating antigen-specific antibodies and specific B- and T-memory cells [167]. The adaptive immune response evolved as a complex regulatory network between the humoral factors and immune cells of both early innate and later adaptive immune mechanisms. The responses start with the presentation of pathogen-derived peptides presented via MHC class I, mainly at peripheral mucosal epithelia, or MHC class II by antigen presenting cells (APC) at peripheral or in systemic lymphoid tissue (Section 3.2). The presentation via MHC I or MHC II determines the adaptive response through Th1 regulation towards cellular, mostly CD8^+^ T-cell, permitted responses or Th2 regulation towards humoral antibody secretion by activated B-cells.

In naïve animals, the survival after infection with virulent RHDV depends in the first instance on the effective innate immune mechanisms simply because of the time delay of between 3 and 7 days until adaptive immune effectors, activated B-cells, or activated cytotoxic T-cells fully functionally respond after the first pathogen recognition. After the infection of juvenile rabbits with virulent RHDV strains, a robust innate immune response prevents completely the development of RHD by limiting the viral replication. However, in adult rabbits, the innate mechanisms mainly fail, and the extremely rapid viral replication leads to severe RHD and high mortality. The different innate immune response in juvenile and adult rabbits involved in resistance and survival against RHD infection then determines a different adaptive immune response. Finally, the immunological memory can differ in response to infection or vaccination with regard to specific T- and B-lymphocytes. This is also seen in rabbits surviving a RHDV infection or after vaccination by the high immunogenicity of RHDV VP1. Overall, the development of adaptive immune responses after infection is always interfered by pathogen-induced counteraction. All of these aspects will be discussed in this paragraph.

#### 3.3.1. Humoral Immunity to RHDV

Calicivirus infections in vertebrate hosts most widely trigger a humoral response, which is involved in infection control through the limitation of virus propagation and the systemic spread of the virus [121,161,168,169]. In rabbits, after RHDV infection, the antibody response is different either between naïve and immune rabbits (survivors of infection or after vaccination), and between juveniles with mostly immature B-cells and adults with a fully mature immune system [6,111,128]. Moreover, the structure of RHDV particles is partially different from other caliciviruses. Mature virions are composed of the VP1 protein building a spherical, non-enveloped icosahedral capsid of 32 to 44 nm. The VP1 that includes the N-terminal arm (NTA), the shell (S), and the protrusion (P) forms in this particle characteristic cup-shaped depressions [170,171,172]. A specific P2 sub-domain of RHDV displays high variability in different RHDV isolates, which contributes to HBGA binding differences. Furthermore, in the most exposed surface loop—L1 (a.a. 300–318)—of VP1, a high variability was detected between different RHDV genotypes. This region contains the neutralizing antibody inducing epitopes [106,107].

As discussed above, in naïve animals the Th2 regulated B-cell response leading to virus-neutralizing antibodies does not provide protection against primary RHDV infection, mainly due to the extremely rapid RHDV replication, the induced T-cell apoptosis, the severe liver destruction by DIC, and due to the time delay between infection and the first appearance of effective anti-RHDV antibody titers (Figure 2). On the other hand, the first RHDV-specific IgM was detectable as early as 50 to 60 h p.i., and is reliably detected at 72 h p.i. [173]. Whether RHDV specific IgM is involved in the decline of the RHDV load in the liver detected at about 50h post infection with virus-capture ELISA [11] is not clear. RHDV particles isolated from rabbit tissues 144 h p.i. were not able to infect naive rabbits, possibly due to the formation of immune complexes or the degradation of viral particles [120].

On the other hand, in surviving rabbits, very high titers of RHDV-neutralizing antibodies not only protect from a severe clinical course and death but also prevent the viral replication in inner organs. Moreover, repeated infection always increases such titers very effectively, which presumably leads to a lifelong immunity [6,11].

The percentage of B lymphocytes (determined as CD19^+^) increased constantly from 8 h to 52 h post infection [99,102,145], followed by high IgM within 2 weeks.

The anti-RHDV IgG which increased slower persisted for months [111]. Interestingly, a repeated vaccination 21 days after the primary one reduced the persistence of protective IgG [111]. The virus-neutralizing capacity of anti-RHDV antibodies was proven by the passive immunization used to stop an RHD outbreak in a rabbit farm [121]. The induction of IgA titers in naïve rabbits which survived an infection was measured in field tests in Australia. Whether these antibodies already prevent infection at peripheral mucosal surfaces is not clear [120].

Three aspects in the antibody response of naïve, juvenile or adult rabbits to a RHDV infection have to be distinguished.

Can naïve young rabbits below 8 weeks of age display a high affinity antibody response despite their mainly immature B-cells? The diversification of the VDJ (variable diversity joining) repertoire seems to start after birth, and it is completely similar to an adult’s by 2–3 months of age [117]. The antibody responsiveness of rabbit kits to immunization also follows the same dynamic. The advantage or disadvantage behind this delayed diversification is not known. However, there are not many diseases that have been reported to affect rabbit kittens, and the resistance to RHDV GI.1 strains is one example of the ability of young rabbits of 4 weeks of age to mount high titers (up to 1.640) after infection [121]. These titers correlated with an increase of B-cells in the liver and spleen, and a robust pro-inflammatory cytokine response. This paradigm changed just recently, when RHDV-2 spread out. However, an involvement of the induced specific antibodies in the immediate clearance of RHDV very early after infection and the protection of the hepatocytes from being infected seems not to be reliable.

Do maternal antibodies from immune mothers influence the antibody response in their offspring? In general, maternal antibodies provide protection from a severe clinical course after infection [174]. Rabbits have a haemochorial placentation, in which maternal antibodies are transmitted from the mother to the offspring through the placenta [175]. In wild rabbit populations, maternal antibodies seem to be important for protection against RHD even after the natural juvenile resistance decline [120,176]. After a single-dose vaccination, RHDV-specific maternal antibodies protected offspring up to the fourth generation. They were mainly transmitted in gestation through trans-placental transmission, and only partly in the lactation period. These maternal antibodies passively protect offspring for at least the first 28 days of life [177]. Specific studies on whether those RHDV-specific maternal antibodies interfere with an active immunization has not yet been studied in detail. However, in general, maternal antibodies attenuate the response to vaccination; especially, the induction of virus-neutralizing antibodies can be reduced. Rabbits have two Cγ gene segments with identical nucleotide sequences, which means that they have only one IgG subclass [178,179], and so no restriction of the IgG subclass is expected during maternal transfer. The stimulation of the immune system of the offspring is not hampered by maternal antibodies, and rabbits with induced pathogen-specific antibodies (IgG + IgA) can be clearly distinguished from rabbits protected from maternal ones (IgG) [120]. The persistence of those antibodies, however, is limited, especially after vaccination [177].

Is the antibody pattern in adult rabbits surviving a RHDV infection different from that of RHDV-resistant juveniles below 8 weeks of age in which a systemic RHDV replication is prevented by a robust innate immunity? Rabbit B cell lymphopoiesis initially occurs in the fetal liver [180,181]: after birth, the bone marrow is the main resource of pre-B-cells, in which they comprise 10–20% of all hematopoietic cells. However, until 8–9 weeks of age, the number of pre-B-cells declines constantly to negligible levels. Moreover, at the age of 4 months, no immature B-cells are present in the bone marrow anymore [182]. However, this time-limited B-lymphopoiesis in bone marrow seems to have no negative impact on the ability of adult rabbits to mount an effective high-affinity antibody response. The general usage of young adult rabbits (6 months of age) for the production of highly-specific antisera indicates the potential of the B-cell repertoire. After RHDV infection, these responses are obviously different in juveniles (<9 weeks of age), with an ongoing B-cell ontogeny from adult rabbits with a fully mature immune system.

#### 3.3.2. Cellular Response

Viral infections are not only controlled by specific virus-neutralizing antibodies in CD4^+^ T-cell-regulated Th2 immune responses but also by CD8^+^ T-cells which, after activation, either lyse virus-infected cells or induce by the secretion of IFNγ a complex network which blocks the viral replication in infected cells without cell destruction. This is very important in metabolic organs like the liver, where massive cell damages lead to immunopathological organ destruction. In RHD, such a process is involved in the fatal outcome in adult rabbits.

The cellular immune responses to RHDV infections are as different as the outcome between juvenile and adult rabbits. In juveniles (<9 weeks of age) no significant changes in the number of T-cells were detected after infection with RHDV GI.1 strains. However, the immunopathology is comparable after infection with GI.2 strains with a remarkable lymphopenia, especially of T-cells [47,49,51,183]. In adults, the apoptosis of lymphocytes is one main pathological process after both RHDV GI.1 and GI.2 strains [6,184]. Especially, the number of CD4(+)Foxp3(+) regulatory T cells (Tregs) decrease in infected adults, whereas in juvenile rabbits the number and frequency of splenic Tregs did not change [50]. Interestingly, the impact on T-cells varies after infection with hemagglutinating or non-hemagglutinating strains. The percentage of all T lymphocytes (CD5^+^), as well as of CD4^+^ T_helper_ cells and CD8^+^ T_cytotoxic_ cells, decreases after infection with HA^-^ strains, but increase slightly after infection with HA^+^, starting 4–8 h p.i. [47,48,144]. In RHDV GI.2 infection, an initial decrease in CD4^+^ and CD8^+^ T-cells is observed, followed by a progressive apoptosis in moribund rabbits, and by recovery and proliferation in surviving adult rabbits. This indicates an involvement of CD8^+^ T-cells in protective responses in naïve adult rabbits [111].

The recruitment of different leukocyte populations to the RHDV-infected liver has been shown to be decisive for the outcome. In adult rabbits, mostly heterophils infiltrate the liver, and appear there near hepatocytes showing severe cellular damage. In contrast, in young rabbits, lymphocytes in close membrane contacts with the cell surface of undamaged hepatocytes dominate [185]. The importance of T-cells in the protection of the liver from severe pathological processes and fatal outcomes after infection was proven in young rabbits that lost their natural resistance after immunosuppressive treatment [125].

### 3.4. Immune Response Pattern Protecting Naïve Rabbits—Lessons to Learn for Vaccination

To summarise the findings about protective immune responses in naïve rabbits against RHDV infection, different patterns are seen in juveniles and adults.

The robust innate immune system based on type I IFN-regulated gene networks limits the RHDV replication already in the periphery, and prevents a systemic infection [125,127,186]. The adaptive immune response, afterwards, is mainly based on neutralizing antibodies, which provide a long-term immunity. In the field, the titer of anti-RHDV antibodies increases strongly after repeated contact with the virus, leading to (presumably lifelong) immunity against homologous or closely-related RHDV strains [187,188]. However, the rapid evolution of RHDV counteracts the antibody-based immunological memory [109,189,190,191,192].

In adults, an extremely rapid RHDV replication overwhelms possible innate responses in the periphery, ending up in a rapid systemic spread and infection of inner organs. The induced Th2 immune responses are not able to reach a level which limits the RHDV-induced pathology, with a fatal outcome. However, as-yet not understood early immune regulatory events direct the immune response to Th1-based mechanisms with the activation of CD8^+^ T-cells, which seem to be able to control the RHDV replication in the liver without destructive cytotoxic activities. In these rare cases, the integrity of the liver is retained, and the infected rabbits survive the infection with less severe clinical signs.

A vaccine, which would trigger the immunoregulatory mechanisms behind these two patterns, might help to eradicate RHDV, because the neutralizing antibody-induced viral evolution is accompanied by cellular immune mechanisms which strictly limit the release of new RHDV variants.

## 4. Vaccines—From Inactivated RHDV-Infected Liver Preparation to Recombinant Vaccines

The extremely high mortality in naïve rabbits after RHDV infection, along with the enormous impact on commercial farmed rabbits and pet rabbits, and finally on wild rabbits in ecosystems lead to high activities in vaccine development. Unfortunately, all of the attempts to establish an in-vitro cell culture system to produce RHDV vaccines failed. Moreover, the emergence of many RHD epizootics and the variability of RHDV increased the need to develop an effective vaccine to limit the expansion of the virus.

### 4.1. Conventional Vaccines against RHDV

Shortly after the emergence of RHDV GI.1, several inactivated RHDV vaccines based on preparations of liver material from RHDV-infected rabbits were developed [54,55,56]. So far, no standard RHDV strain for vaccine development has been established. Therefore, usually the most virulent, local RHDV strains are selected as vaccine strains. All of these inactivated vaccine candidates were tested in ‘classical vaccination-challenge trials’, and the high immunogenicity of inactivated RHDV was proven. In these studies, the induction of virus-neutralizing antibodies was shown, lasting mostly for more than 12 months post vaccination. All of these vaccines not only prevent severe clinical courses of RHD and high mortality but also inhibit RHDV replication [54,56]. However, after the emergence of RHDV2 in 2010, new vaccines had to be developed, as these classical vaccines were shown to be very effective in inducing long term protection against homologous RHDV GI.1 strains, but provided only limited cross protection against GI.2 genogroup strains [193]. The shortly-thereafter developed liver-derived RHDV2 vaccines (FILAVAC VHD K C+V, Filavie S.A.S; ERAVAC, Hipra) also conveyed only limited cross protection to other RHDV strains, so that rabbits need to be vaccinated either with two different vaccines or with polyvalent vaccines, leading to higher economical efforts for breeders. Moreover, the observed variability of RHDV with the potential emergence of further virus variants showed the limitations of a vaccine development strategy based on liver preparations from infected rabbits.

### 4.2. Recombinant Vaccines against RHDV

In order to improve vaccine production methods using the liver material of infected rabbits, numerous alternative approaches were developed in the last few years. However, the induction of general protection against RHDV is not the sole focus. Several claims have been made in regard to a recombinant vaccine. A vaccine should be ethically justifiable, induce protection shortly after administration, induce long term protection, convey cross protection against several RHDV strains, be economical, and ideally be administered only once during a rabbit’s life.

In order to achieve some or all of these goals, since the 1990s, many recombinant vaccines against RHDV have been developed [58,59,194]. In most cases, the focus was on the expression of the capsid protein VP1, as it was shown to be very immunogenic [58,105]. Recombinant VP1 was expressed using different expression vectors, such as baculovirus [58,111,128,194], adenovirus [195], myxoma virus [60], yeast [196], Escherichia coli [197], vaccinia virus [61], insect cells [59], or plants [198].

One desired effect of VP1 expression is the spontaneous formation of virus-like particles (VLP). Because of their similarity to infectious virus particles, they have more immunogenic potential than non-assembled VP1, and therefore can elicit a strong humoral and cellular immune response [199,200,201]. VLP are taken up by antigen-presenting cells, and are presented by MHC class II molecules, leading to the activation of DC, cytokine release, and CD4^+^ T-cell activation, as well as by MHC class I molecules, leading to CD8^+^ T-cell activation [200,202]. Due to their small size, VLP can spread to lymph nodes to interact with even more CD4^+^ T-cells. In addition to a potent cellular immune response, a humoral immune response is also triggered by the activation of B-cells, leading to high antibody titers and the generation of memory B-cells [200]. In the few last years, vaccines based on VP1 VLP have therefore been brought into focus [111,128,197,203].

So far, to our knowledge, only two recombinant vaccines have come to market (NOBIVAC^®^ Myxo-RHD PLUS + NOBIVAC^®^ Myxo-RHD, MSD Animal Health), with only one conveying protection also against RHDV2. Both are based on a myxoma virus vector expressing RHDV/RHDV2 VP1.

### 4.3. New Approaches in Vaccine Design

Classical vaccine development approaches concentrate on the administration of antigens, which are ideally very immunogenic, in order to induce a specific and hopefully long-lasting immune response and therefore protection against the pathogen. With many vaccines, however, the duration of protection is often limited to only a few years at most. Additionally, in order to prolong protection, adjuvants are needed for the enhancement of the immune response. Furthermore, vaccines need to be boosted over the years in order to maintain protection. Classical vaccines are often developed against ‘easy pathogens’ based on the use of the structural proteins of those pathogens, which are known for high immunogenicity. However, the challenges of modern vaccine development lie more with pathogens that hide from the immune system, or which change and mutate constantly, such as HIV, Malaria, and Ebola, etc. [204]. In order to gather more potential immunomodulatory protein candidates, a novel process—reverse vaccinology—was developed 20 years ago. Whole virus and bacteria genomes are scanned, and immunological epitopes are predicted. Furthermore, conserved epitopes in core proteins are predicted to develop vaccines against many related strains of pathogens [205,206]. One example for reverse vaccinology is the development of a vaccine candidate against Ebola. With epitope prediction tools, proteins containing epitopes which present to human leukocyte antigen molecules and induce IFNγ in peripheral blood mononuclear cells were found [207]. In the last decade, a new discipline came into focus: systems vaccinology. It combines reverse vaccinology, which studies the role of immunogens, with additional factors like adjuvants and antigen delivery. With certain data-sets gained from systems-based studies, mainly by -omics techniques, it is possible to develop better vaccines based on improved insight into the immune response induced by the vaccine candidate, and to detect correlates of protection [204].

As for RHDV vaccine development, the fact that, in recent years, another highly-pathogenic virus variant emerged, which left the classic vaccines useless, demonstrated that it is important to constantly improve vaccines by, for example, considering the antigenicity of the core proteins of different virus strains to protect against possible new emerging strains, and to strive for not only B-cell mediated immunity but also adapted cellular and trained immunity, so that vaccines induce a full, long lasting protection.

## Figures and Tables

**Figure 1 vaccines-09-00255-f001:**
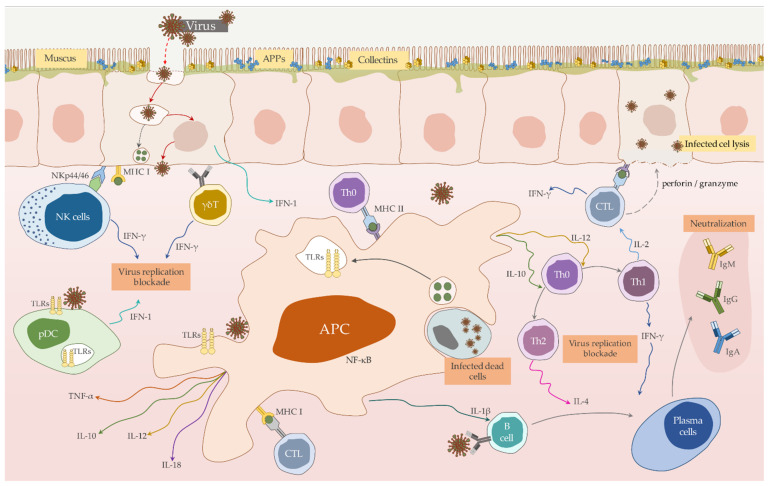
Schematic representation the mutual interaction of immune cells proven or assumed to be involved in the immune response against *Lagovirus europaeus* infection.

**Figure 2 vaccines-09-00255-f002:**
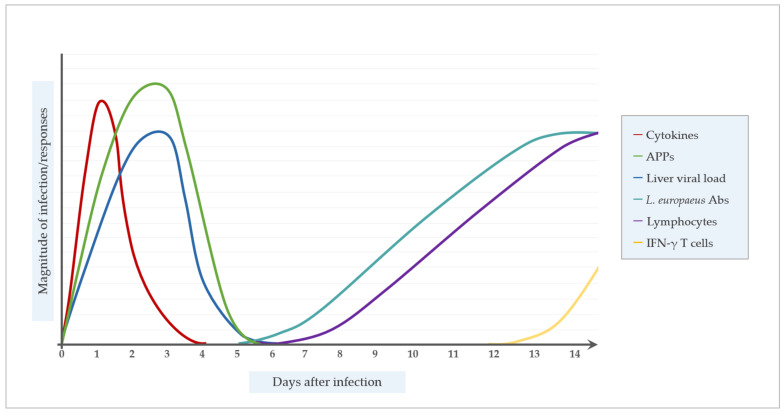
Temporal dynamic of the immune response after *Lagovirus europaeus* infection.; APPs, acute phase proteins; *L. europaeus* Abs, *Lagovirus europaeus* antibodies.

**Table 1 vaccines-09-00255-t001:** Overview of liver-derived vaccines against RHDV, licensed in Germany (PEI, state 11.01.2021).

Vaccine	Containing RHDV Strains	Manufacturer	Date of Admission	Admission Number
RIKA-VACC RHD	RHDV GI.1 strain ‘Eisenhüttenstadt’ inactivated	Ecuphar NV	04.09.2003	200a/91
RIKA-VACC Duo	myxoma virus strain ‘CAMP V-219’ attenuated RHDV GI.1 strain ‘CAMP V-351’ inactivated	Ecuphar NV	12.06.2008	PEI.V.03071.01.1
CUNIVAK RHD	RHDV GI.1 strain ‘Eisenhüttenstadt’ inactivated	Ceva Tiergesundheit GmbH	11.05.2004	206a/92
CUNIVAK COMBO	myxoma virus strain ‘CAMP V-219’ attenuated	Ceva Tiergesundheit GmbH	05.08.2009	PEI.V.07962.01.1
	RHDV GI.1 strain ‘CAMP V-351’ inactivated			
Eravac	RHDV GI.2 strain ‘V-1037’ inactivated	Laboratorios Hipra S.A.	26.09.2016	EU/2/16/199
Filavac VHD K C+V	RHDV GI.1 strain ‘IM507.SC.2011’ inactivated	FILAVIE, Roussay	13.03.2017	PEI.V.11900.01.1
	RHDV GI.2 strain ‘LP.SV.2012’ inactivated			

**Table 2 vaccines-09-00255-t002:** Comparison between the main types of *Lagovirus europaeus*.

Genotype	GI.1; RHDV1	GI.1a; RHDV1a	GI.2; RHDV2
Place/year of detection	China, 1984	Italy, 1997	France, 2010
Infected species	*O. cuniculus*	*O. cuniculus*	*O. cuniculus* several *Lepus species*
Average mortality rate	>90%	≤100%	>90%
Disease symptoms	Typical for RHD	Typical for RHD	Typical for RHD
Juvenile resistance	until 9 weeks		until 4 weeks
Cross protective in surviving animals	limited for RHDV2	No	No

## Data Availability

Not applicable.

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
