# Peer review of "Immunity against Lagovirus europaeus and the Impact of the Immunological Studies on Vaccination"

_vaccines, 2021, doi:10.3390/vaccines9030255_

Round 1
Reviewer 1 Report
The manuscript entitled “Immunity against Lagovirus europaeus and the impact of immunological studies on vaccination” is important and provides a very relevant review in the immunity and vaccination against lagovirus. I think that the paper is adequate to Vaccines. However, the paper could be substantial improved if the authors made the following changes:
Abstract
“However, juvenile rabbits (< 9 weeks of age) with an immature immune system display a general resistance, whereas less than 3% of adult rabbits survive an infection”. This is observed for RHDV but not for RHDV2. The authors should rewrite it and add information for RHDV2.
Introduction
“RHDV had also a negative impact in Mediterranean ecosystems, where rabbits are a food base for endangered endemic species, including the Iberian eagle (Aquila adalberti) 53 and Iberian lynx (Lynx pardinus) [17]”. The reference 17 is not proper, please replace by Monterroso et al, 2016 Scientific Report and Rouco et al., 2018 TBED.
“Until 2010 all RHD outbreaks were caused by the so-called ‘classical’ RHDV designated as RHDV1”. Please rephrase the sentence; these strains were never called RHDV1, the different strains were classified in G1-G6 genogroups (Le Gall et al, 1998, J General Virology)
“In 2010 in France a new RHDV strain emerged, it was designated as RHDV2 which not only infects O. cuniculus, but also hare species, Lepus capensis mediterraneus, L. corsicanus, L. europaeus, L. timidus and L. granatensis [27]”. Please change the sentence. The RHDV2 was described in the reference 27 but the presence of RHDV2 in Lepus Lepus capensis mediterraneus, L. corsicanus, L. europaeus, L. timidus was described latter by Neimanis et al 2018 BMC Vet Res; Le Gall-Reculé et al, 2017 Vet Research; Velarde et al, 2017 TBED; Camarda et al, 2014, Res Vet Sci; Puggioni et al, 2013, Vet Res. For Lepus granatensis it was reported the infection with RHDV and not RHDV2 (Lopes et al., 2014, Vet Res).
Before to start talk about the GI.1 and GI.2 strains, the authors should add a paragraph explaining what is the GI.1 and GI.2 strains (see Le Pendu et al., 2017, J General Virol).
“One problem of all vaccines is the variability of RHDV GІ.1 and GІ.2 strains based on their high recombination ability with limited cross-protective potential [54] of RHDV GІ.1 and GІ.2 strains based on their high recombination ability with limited cross-protective potential [54]”. The authors should explain that two different types of recombination have been described in RHDV virus. Recombination in the capsid (Abrantes et al., 2008 Archives of Virology; Forrester et al 2007 Virology; Hu et al. 2017 Archives of Virology) and recombination between the RHDV structural and non-structural regions (Lopes et al 2015 J Gen Virol. and reference 54)
The origin of RHDV is not fully understood. The pathogenic virus forms may have evolved from avirulent strains which circulated in European rabbits asymptomatically [56-60] at least 30 years before the first outbreak of RHD in China. Another hypothesis suggested by Esteves et al., 2017 Plos Pathogens should also be included. In this alternative hypothesis the authors suggest that the RHDV arose from a species jump from species sympatric with European leporids, either native or previously introduced like American rabbit (Sylvilagus).
Lagovirus europaeus belongs to the Caliciviridae family, genus Lagovirus. Add the reference Vinje et al., 2019 J general Virology
“It has been divided into two main gene groups associated with RHDV (GI) or with the European brown hare syndrome virus (EBHSV, GII)”. Rephrase as “Le Pendu et al., 2017 J gen virology renamed the RHDV nomenclature and the lagovirus strains were divided into two main gene groups associated with RHDV (GI) or with the European brown hare syndrome virus (EBHSV, GII)”.
The EBHSV GII genogroup contains two genotypes: Rephrase as the EBHSV can be divided into two groups (Lopes et al., 2014 Virology) These two groups were classified as Le Pendu et al, 2017 as GII genogroup that contains two genotypes:…..
Missing virus receptor in peripheral mucosal surfaces
Please discuss in more detail the role of Histo-blood groups in the different RHDV strain infection (see Nystron et al., 2011, Plos Pathogens and Lopes et al., 2018 J Virology)
Antigen presentation
It has been reported that the CD4 gene show an overall higher divergence in lagomorphs compared to primates with highest divergence in the D2 domain (de Sousa-Pereira et al., 2016, Immunogenetics. Add this information and include in the discussion.
Explain what do you mean as immature B-cells in young rabbit. Do you mean absence of VDJ repertoire diversification? How can these be an advantage for the young rabbits?
The leporids are unique because they have only one IgG that evolved under positive selection (Pinheiro et al. 2014 Open Biology) and they have at least 14 IgA (Pinheiro et al., 2018 Plos One) make a comment how these can be an advantage or disadvantage to build an efficient immune response.
The rabbit is a good model to study human diseases and for diagnostic and therapeutics development (Esteves et al., 2018 Exp Mol Med; Mage et al, 2019 Dev Comp Immunol.). Please make a statement how the study of RHDV in young and adult rabbits could be useful in this context.
Author Response
Dear Reviewer,
On behalf of the authors of the article "Immunity against Lagovirus europaeus and the impact of the immunological studies on vaccination", written by Claudia Müller, RafaÅ‚ Hrynkiewicz, Dominika BÄ™bnowska, Jaime Maldonado, Massimiliano Baratelli, Bernd Köllner and me, we would like to thank you for the informative and detailed review of our article. We believe that your the excellent knowledge and commitment influenced our article and made it much better. We followed your suggestions and all changes are marked or highlighted in the updated version of the manuscript. Here are the point-by-point answers.
- Abstract:
- “However, juvenile rabbits (< 9 weeks of age) with an immature immune system display a general resistance, whereas less than 3% of adult rabbits survive an infection”. This is observed for RHDV but not for RHDV2. The authors should rewrite it and add information for RHDV2.
RE: As recommended by the reviewers, we analyzed the abstract and added mortality information for RHDV2.
- Introduction:
- “RHDV had also a negative impact in Mediterranean ecosystems, where rabbits are a food base for endangered endemic species, including the Iberian eagle (Aquila adalberti) 53 and Iberian lynx (Lynx pardinus) [17]”. The reference 17 is not proper, please replace by Monterroso et al, 2016 Scientific Report and Rouco et al., 2018 TBED.
RE: As suggested by the reviewer, we improved the ref. [17]. We added two new references: Monterroso et al, 2016; Rouco et al., 2018.
- “Until 2010 all RHD outbreaks were caused by the so-called ‘classical’ RHDV designated as RHDV1”. Please rephrase the sentence; these strains were never called RHDV1, the different strains were classified in G1-G6 genogroups (Le Gall et al, 1998, J General Virology).
RE: We analyzed the reviewer's suggestions and decided to rearrange the sentence.
- “In 2010 in France a new RHDV strain emerged, it was designated as RHDV2 which not only infects O. cuniculus, but also hare species, Lepus capensis mediterraneus, L. corsicanus, L. europaeus, L. timidus and L. granatensis [27]”. Please change the sentence. The RHDV2 was described in the reference 27 but the presence of RHDV2 in Lepus Lepus capensis mediterraneus, L. corsicanus, L. europaeus, L. timidus was described latter by Neimanis et al 2018 BMC Vet Res; Le Gall-Reculé et al, 2017 Vet Research; Velarde et al, 2017 TBED; Camarda et al, 2014, Res Vet Sci; Puggioni et al, 2013, Vet Res. For Lepus granatensis it was reported the infection with RHDV and not RHDV2 (Lopes et al., 2014, Vet Res).
RE: We improved the literature and information on L. granatensis.
- Before to start talk about the GI.1 and GI.2 strains, the authors should add a paragraph explaining what is the GI.1 and GI.2 strains (see Le Pendu et al., 2017, J General Virol).
RE: After analyzing the reviewer's suggestions, we decided to add RHDV (GI.1) and RHDV2 (GI.2). A more detailed description of GI.1 and GI.2 is discussed in Chapter 2 (Lagovirus europaeus - epidemiological history of RHD causative agent).
- “One problem of all vaccines is the variability of RHDV GІ.1 and GІ.2 strains based on their high recombination ability with limited cross-protective potential [54] of RHDV GІ.1 and GІ.2 strains based on their high recombination ability with limited cross-protective potential [54]”. The authors should explain that two different types of recombination have been described in RHDV virus. Recombination in the capsid (Abrantes et al., 2008 Archives of Virology; Forrester et al 2007 Virology; Hu et al. 2017 Archives of Virology) and recombination between the RHDV structural and non-structural regions (Lopes et al 2015 J Gen Virol. and reference 54).
RE: As recommended by the reviewer, we have updated our review with information on RHDV recombination.
- The origin of RHDV is not fully understood. The pathogenic virus forms may have evolved from avirulent strains which circulated in European rabbits asymptomatically [56-60] at least 30 years before the first outbreak of RHD in China. Another hypothesis suggested by Esteves et al., 2017 Plos Pathogens should also be included. In this alternative hypothesis the authors suggest that the RHDV arose from a species jump from species sympatric with European leporids, either native or previously introduced like American rabbit (Sylvilagus).
RE: As suggested by the reviewer, we have added information about the second hypothesis of the RHDV development.
- Lagovirus europaeus belongs to the Caliciviridae family, genus Lagovirus. Add the reference Vinje et al., 2019 J general Virology.
RE: As recommended by the reviewer, we added ref. 91 (Vinjé, J. et al., 2019)
- “It has been divided into two main gene groups associated with RHDV (GI) or with the European brown hare syndrome virus (EBHSV, GII)”. Rephrase as “Le Pendu et al., 2017 J gen virology renamed the RHDV nomenclature and the lagovirus strains were divided into two main gene groups associated with RHDV (GI) or with the European brown hare syndrome virus (EBHSV, GII)”.
RE: As recommended by the reviewer, we corrected the sentence from "It has been divided into two main gene groups associated with RHDV (GI) or with the European brown hare syndrome virus (EBHSV, GII)" to the sentence "Le Pendu et al. renamed the RHDV nomenclature and the lagovirus strains were divided into two main gene groups associated with RHDV (GI) or with the European brown hare syndrome virus (EBHSV, GII) ".
- The EBHSV GII genogroup contains two genotypes: Rephrase as the EBHSV can be divided into two groups (Lopes et al., 2014 Virology) These two groups were classified as Le Pendu et al, 2017 as GII genogroup that contains two genotypes:…..
RE: As recommended by the reviewer, we decided to change the sentence from "The EBHSV GII genogroup contains two genotypes:" per sentence "The EBHSV can be divided into two groups (Lopes et al., 2014 Virology). These two groups were classified as Le Pendu et al., as GII genogroup that contains two genotypes:".
- Missing virus receptor in peripheral mucosal surfaces
- Please discuss in more detail the role of Histo-blood groups in the different RHDV strain infection (see Nystron et al., 2011, Plos Pathogens and Lopes et al., 2018 J Virology).
RE: In our opinion, there is no connection between HBGA and the immune response of the hosts against RHDV in the existing literature. HBGA is the cellular receptor which allow the virus entry; the variability of the receptor and the affinity with RHDV are topics more related with virology rather than immunology.
- Antigen presentation
- It has been reported that the CD4 gene show an overall higher divergence in lagomorphs compared to primates with highest divergence in the D2 domain (de Sousa-Pereira et al., 2016, Immunogenetics. Add this information and include in the discussion.
RE: The data has been added, thank you for this valuable comment.
- Explain what do you mean as immature B-cells in young rabbit. Do you mean absence of VDJ repertoire diversification? How can these be an advantage for the young rabbits?
RE: we do agree with the Reviewer, that the diversification of the VDJ repertoire seems to start after birth and it is completely similar to adult by 2-3 months of age (Knight KL, Crane MA. Generating the antibody repertoire in rabbit. Adv Immunol. 1994;56:179-218. doi: 10.1016/s0065-2776(08)60452-6. PMID: 8073947). The antibody responsiveness of rabbit kits to immunization also follow the same dynamic. This characteristic is specific of rabbits but also pigs.
In our opinion, the advantage or disadvantage behind this delayed diversification is not known. Despite this, it is interesting to highlight that there are not many diseases which affect rabbit kits. For example, rabbit kits are known to be resistant to RHDV whereas adult rabbit are susceptible. This paradigm changed just recently when RHDV-2 spread out. May be there has been no need for rabbit to evolve in such way because they relies mostly on the innate and maternal immunity rather than on an adaptive antibody response.
- The leporids are unique because they have only one IgG that evolved under positive selection (Pinheiro et al. 2014 Open Biology) and they have at least 14 IgA (Pinheiro et al., 2018 Plos One) make a comment how these can be an advantage or disadvantage to build an efficient immune response.
RE: In other species but rabbit the IgG diversifies in terms of structure and functions. Selective subclass deficiencies are usually not detrimental to the individual but do sometimes lead to enhanced susceptibility toward specific classes of pathogens. Rabbits are susceptible to several classes of pathogen and immunization (when the animal survive to the infection or is vaccinated) provide protection for several of them as other species. Therefore, the advantage or disadvantage of such lack of diversity in term of protection are unknown, as far as we know. Unfortunately, we are not aware of study meant to answer such questions or why the immune system of rabbit has evolved in such way.
- The rabbit is a good model to study human diseases and for diagnostic and therapeutics development (Esteves et al., 2018 Exp Mol Med; Mage et al, 2019 Dev Comp Immunol.). Please make a statement how the study of RHDV in young and adult rabbits could be useful in this context.
RE: RHD has been used as experimental model to test novel therapeutic approaches against acute liver failure (ALF) in humans:
- Tuñon MJ, San Miguel B, Crespo I, et al. Cardiotrophin-1 promotes a high survival rate in rabbits with lethal fulminant hepatitis of viral origin. J Virol. 2011;85(24):13124-13132. doi:10.1128/JVI.05725-11.
- Tuñón MJ, San-Miguel B, Crespo I, Laliena A, Vallejo D, Álvarez M, Prieto J, González-Gallego J. Melatonin treatment reduces endoplasmic reticulum stress and modulates the unfolded protein response in rabbits with lethal fulminant hepatitis of viral origin. J Pineal Res. 2013 Oct;55(3):221-8. doi: 10.1111/jpi.12063. Epub 2013 May 16. PMID: 23679826.
It has been observed, that the disease produced by the virus in rabbits resemble to the human ALF from clinical point of view. Moreover, several other clinical aspects of RHD, like the production of disseminated intravascular coagulation (DIC) or coagulopathy resemble to those produced by other viral disease in humans like those caused by Influenza A virus or SARS-CoV-2. In these, the clinical outcome has been suggested to be caused by a previous cytokine storm. The reproduction of such clinical sign in animal models is not an easy task as normally these animals are not the normal hosts of the virus (e.g. mouse) neither the predisposing factors are completely known.
In RHD, these clinical signs can be easily reproduced though they have been likely associated to the lesions caused to the liver of the rabbit (Trzeciak-Ryczek A, Tokarz-Deptuła B, Deptuła W. The importance of liver lesions and changes to biochemical and coagulation factors in the pathogenesis of RHD. Acta Biochim Pol. 2015;62(2):169-71. doi: 10.18388/abp.2014_943. Epub 2015 Apr 23. PMID: 25918886.). Despite this their association with previous cytokine storm as not been investigated. This aspect should be deeply addressed in future studies as the RHD might be a model of viral produced immune pathology.
Since the data concerns the pathogenesis of the disease rather than the immune response, in our opinion it does not suit the paper it its current form.
We hope that our additions improved our manuscript and will fullfil your expectations.
Kind regards,
Paulina Niedźwiedzka-Rystwej
Reviewer 2 Report
This is a major review and should be generally useful as it brings together field epidemiology as well as work on the development of vaccines with a wider understanding of the mechanisms that underlie the rabbits immune defenses. Nonetheless, I have a couple of general comments:
I believe it would be useful to present the work more factually as an example of co-evolution between virus and host rather than implying that the virus had to 'learn' to use host cell machinery. This would enable readers to see virulence and resistance as a dynamic balance or an 'arms race' where increases in resistance are met with increases in virus virulence. This best explains why RHDV in the field still kills over 70% of infected rabbits even after circulating for 30 years. See for example papers by;
Elsworth, P., Cooke, B.D., Kovaliski, J. Sinclair, R., Holmes, E.C. and Strive, T. (2014). Increased virulence of rabbit haemorrhagic disease virus associated with genetic resistance in wild Australian rabbits (Oryctolagus cuniculus). Virology, 464 – 465, 415-423 https://doi.org/10.1016/j.virol.2014.06.037
Elsworth, P.G., Kovaliski, J. and Cooke, B. (2012). Rabbit Haemorrhagic Disease: Are Australian rabbits (Oryctolagus cuniculus) evolving resistance to infection with Czech CAPM 351 RHDV? Epidemiology and Infection. 140, 1972-81 doi: 10.1017/S0950268811002743
Di Giallonardo F, Holmes EC. Viral biocontrol: grand experiments in disease emergence and evolution. Trends Microbiol. 2015 Feb;23(2):83-90. doi: 10.1016/j.tim.2014.10.004. Epub 2014 Oct 31. PMID: 25455418; PMCID: PMC4323752.
I also think the readership of this review could be advantageously extended and brought up-to-date with information from southern USA and northern Mexico where RHDV2 has recently spread through native lagomorphs (cottontails, jackrabbits). Both countries may gain from knowing more about vaccines if they wish to conserve native species. Much of this is still in the 'grey' literature and has been under-reported because Covid-19 has prevented serious field work. However a Google search will bring up newspaper reports and reports from the Center for Disease Control etc.
Beyond those points, I think there is room for improvement in the English grammar. To give a few examples:
Abstract, line 18, 'was finally leading to' would be better as 'finally led to'
Similarly in line 20, 'is finally leading to an' could be simplified to 'resulted in'
Line 57, 'the causing agent' would be better as the 'causative' agent.
Line 106, Heading 'RHD causing agent' I think RHDV could be used here as it is previously defined as the causative agent.
I do not have the time to go through the whole manuscript like this but if possible, I would advise that correction of similar minor points by a native English speaker would give this manuscript some extra shine, adding to its value.
Author Response
Dear Reviewer,
On behalf of the authors of the article "Immunity against Lagovirus europaeus and the impact of the immunological studies on vaccination", written by Claudia Müller, RafaÅ‚ Hrynkiewicz, Dominika BÄ™bnowska, Jaime Maldonado, Massimiliano Baratelli, Bernd Köllner and me, we would like to thank you for the informative and detailed review of our article. We believe that your excellent knowledge and commitment influenced our article and made it much better. We followed the suggestions, and all changes are marked or highlighted in the updated version of the manuscript. Here are the point-by-point answers.
- I believe it would be useful to present the work more factually as an example of co-evolution between virus and host rather than implying that the virus had to 'learn' to use host cell machinery. This would enable readers to see virulence and resistance as a dynamic balance or an 'arms race' where increases in resistance are met with increases in virus virulence. This best explains why RHDV in the field still kills over 70% of infected rabbits even after circulating for 30 years. See for example papers by;
Elsworth, P., Cooke, B.D., Kovaliski, J. Sinclair, R., Holmes, E.C. and Strive, T. (2014). Increased virulence of rabbit haemorrhagic disease virus associated with genetic resistance in wild Australian rabbits (Oryctolagus cuniculus). Virology, 464 – 465, 415-423 https://doi.org/10.1016/j.virol.2014.06.037
Elsworth, P.G., Kovaliski, J. and Cooke, B. (2012). Rabbit Haemorrhagic Disease: Are Australian rabbits (Oryctolagus cuniculus) evolving resistance to infection with Czech CAPM 351 RHDV? Epidemiology and Infection. 140, 1972-81 doi: 10.1017/S0950268811002743
Di Giallonardo F, Holmes EC. Viral biocontrol: grand experiments in disease emergence and evolution. Trends Microbiol. 2015 Feb;23(2):83-90. doi: 10.1016/j.tim.2014.10.004. Epub 2014 Oct 31. PMID: 25455418; PMCID: PMC4323752.
RE: The problem of the relationship between the host and the rabbit haemorrhagic disease virus has been explained as an example of an „arms run”, in which the virus uses its own mechanisms to maintain a high level of virulence due to the increased resistance of the rabbit. The references proposed by the Reviewer have been added.
- I also think the readership of this review could be advantageously extended and brought up-to-date with information from southern USA and northern Mexico where RHDV2 has recently spread through native lagomorphs (cottontails, jackrabbits). Both countries may gain from knowing more about vaccines if they wish to conserve native species. Much of this is still in the 'grey' literature and has been under-reported because Covid-19 has prevented serious field work. However a Google search will bring up newspaper reports and reports from the Center for Disease Control etc.
RE: The situation of the United States and Mexico related to the spread of RHDV2 from 2020 has been characterized.
- Beyond those points, I think there is room for improvement in the English grammar. To give a few examples:
- Abstract, line 18, 'was finally leading to' would be better as 'finally led to'
- Similarly in line 20, 'is finally leading to an' could be simplified to 'resulted in'
- Line 57, 'the causing agent' would be better as the 'causative' agent.
- Line 106, Heading 'RHD causing agent' I think RHDV could be used here as it is previously defined as the causative agent.
RE: Grammar errors have been corrected, thank you for the comment.
- I do not have the time to go through the whole manuscript like this but if possible, I would advise that correction of similar minor points by a native English speaker would give this manuscript some extra shine, adding to its value.
RE: Thank you for your valuable tip. As recommended, the text has been corrected by a native English speaker.
Again, we would like to thank you and we are hoping to fullfill your expectations.
Kind regards,
Paulina Niedźwiedzka-Rystweh
Reviewer 3 Report
The review by Müller et al provides a comprehensive overview on the pathology of Rabbit hemorrhagic disease virus (RHDV) and thereby, the implications for vaccines against it. The sections are well organized and structured. I have a few minor comments:
1. Interestingly, the authors mention that an increase of the activity of TNF-α, TGF-β and IL-1β was observed in the liver whereas the analysis of cytokine patterns in peripheral blood leukocytes showed a decrease in IL-1β, -2, -18 activity. Is there anything known about the role of other cell death pathways such as inflammasomes (pyroptosis) in the pathology? What is the source of IL-1β since apoptosis does not lead to bursting of the cell and leakage of cellular contents?
2. Also, can the authors elaborate the role of plasmacytoid dendritic cells in RHDV since the authors show these cells to be a source of IFN-1 in the schematic?
3. Are there published studies that mechanistically dissect how IFN-1 blocks viral replication? e.g. which stage of the viral replication is affected?
Author Response
Dear Reviewer,
On behalf of the authors of the article "Immunity against Lagovirus europaeus and the impact of the immunological studies on vaccination", written by Claudia Müller, RafaÅ‚ Hrynkiewicz, Dominika BÄ™bnowska, Jaime Maldonado, Massimiliano Baratelli, Bernd Köllner and me, we would like to thank you for the informative and detailed review of our article. We believe that your excellent knowledge and commitment influenced our article and made it much better. We followed your suggestions of all Reviewers and tried to fulfill them, and all changes are marked or highlighted in the updated version of the manuscript. Here are the point-by-point answers.
- Interestingly, the authors mention that an increase of the activity of TNF-α, TGF-β and IL-1β was observed in the liver whereas the analysis of cytokine patterns in peripheral blood leukocytes showed a decrease in IL-1β, -2, -18 activity. Is there anything known about the role of other cell death pathways such as inflammasomes (pyroptosis) in the pathology? What is the source of IL-1β since apoptosis does not lead to bursting of the cell and leakage of cellular contents?
RE: Several studies indicated that apoptosis of different leukocyte populations (T-cells; monocytes) is directly correlated to the pathological process in the very rapid disease progression, especially in adult rabbits. This is on topic in this review. Whether or not pyroptosis as cell death pathway is induced was not yet investigated and therefore not discussed in this review.
- Also, can the authors elaborate the role of plasmacytoid dendritic cells in RHDV since the authors show these cells to be a source of IFN-1 in the schematic?
RE: In the past 20 years several studies proofed that plasmacytoid dendritic cells (pDC) which are rare cells in blood and lymphoid organs, are the most potent producers of type I and type III IFNs in inner organs in response enveloped viruses sensed by TLR7 or TLR9, respectively (Izaguirre et al., 2003; Coccia et al., 2004; Ito et al., 2006; Li et al., 2018; etc.).
Izaguirre A, Barnes BJ, Amrute S, Yeow WS, Megjugorac N, Dai J, et al. Comparative analysis of IRF and IFN-alpha expression in human plasmacytoid and monocyte-derived dendritic cells. J Leukoc Biol 2003;74:1125–38.
Coccia EM, Severa M, Giacomini E, Monneron D, Remoli ME, Julkunen I, et al. Viral infection and Toll-like receptor agonists induce a differential expression of type I and lambda interferons in human plasmacytoid and monocyte-derived dendritic cells. Eur J Immunol 2004;34:796–805.
Ito T, Kanzler H, Duramad O, Cao W, Liu YJ. Specialization, kinetics, and repertoire of type 1 interferon responses by human plasmacytoid predendritic cells. Blood 2006;107:2423–31.
Li S., · Gong M. · Zhao F. · Shao J. · Xie Y. · Zhang Y. · Chang H. Type I Interferons: Distinct Biological Activities and Current Applications for Viral Infection. Cell Physiol Biochem 2018;51:2377–2396
Matthew J. Neave, Robyn N. Hall, Nina Huang, Kenneth A. McColl, Peter Kerr, Marion Hoehn, Jennifer Taylor, and Tanja Strive. Robust Innate Immunity of Young Rabbits Mediates Resistance to Rabbit Hemorrhagic Disease Caused by Lagovirus Europaeus GI.1 But Not GI.2 Viruses. 2018 Sep; 10(9): 512.
Interestingly, the direct role of pDC in response to RHDV is not well investigates so far. The lack of appropriate tools to distinguish this dendritic cell population is one main reason for it. However, the transcriptome studies in RHDV resistance phenomena in juvenile rabbits showed the importance of an early Type I; II and III response especially in the liver (Neave et al., 2018).
To more clearly differentiate was is known from what can be assumed we changed the figure legend to:
Figure 1. Schematic representation their mutual interaction of immune cells proofed or assumed to be involved in the immune response against of Lagovirus europaeus infection.
- Are there published studies that mechanistically dissect how IFN-1 blocks viral replication? e.g. which stage of the viral replication is affected?
RE: The general pathways in which IFN-1 is inducing IFN induced genes which than are involved in blocking viral replication is in detail characterized for several viral families, including Caliciviridae and reviewed in detail a recent comprehensive review (Penflor-Tellez et al., 2019 Frontiers Immunol., 10:2334).
Yoatzin Peñaflor-Téllez , Adrian Trujillo-Uscanga , Jesús Alejandro Escobar-Almazán , Ana Lorena Gutiérrez-Escolano Immune Response Modulation by Caliciviruses. Front Immunol. 2019 Oct 1;10:2334. doi: 10.3389/fimmu.2019.02334.
This is not investigated yet in detail for RHDV and the present review focus on immune mechanisms which prevent the fatal outcome after RHDV infection and what one can be learn for rational vaccine design. Therefore, this is not discussed in detail here.
Again, we would like to thank you for your effort and time and we are hoping that our manuscript in its current for will fulfill the requirements of the Vaccines journal.
Thank you for your time and consideration,
On behalf of the Authors,
Paulina Niedźwiedzka-Rystwej
Round 2
Reviewer 1 Report
The authors did a good job and improved the paper.
I have only a minor correction:
In 2014, both classic RHDVGI.1 strains and RHDV GI.2 strains were 66 detected in L. granatensis, indicating that bothnhave the ability to break the species barrier and infect other species besides O. cuniculus [30-35]. This information is wrong. The authors should write that “In 1990s, the classic RHDVGI.1 strains were detected in L. granatensis [35]. These infections in Lepus species indicates that both RHDV GI.1 and GI.2 have the ability to break the species barrier and infect other species besides O. cuniculus [30-35]”. In fact, RHDV GI.2 was not yet detected in L. granatensis.
Author Response
Dear Reviewer,
Thank you for checking our paper and positive feedback on our corrections. As far as the minor concern of yours is concerned, we absolutely agree with your statement that our sentence: "In 2014, both classic RHDVGI.1 strains and RHDV GI.2 strains were 66 detected in L. granatensis, indicating that both have the ability to break the species barrier and infect other species besides O. cuniculus [30-35]" includes wrong information, and therefore we have changed the sentence using your suggestion. The change is marked in the current version of the manuscript by a grey highlight.
Once again, thank you for your time and consideration,
On behalf of the Authors,
Paulina Niedźwiedzka-Rystwej